# Microstructure and Texture Evolution of the Magnesium Alloy ZMX210 during Rolling and Annealing

**DOI:** 10.3390/ma16124227

**Published:** 2023-06-07

**Authors:** Gerrit Kurz, Ketan Ganne, Maria Nienaber, Jan Bohlen

**Affiliations:** Institute of Material and Process Design, Helmholtz-Zentrum Hereon GmbH, Max-Planck-Straße 1, 21502 Geesthacht, Germany; gketan.balaji.met14@iitbhu.ac.in (K.G.); maria.nienaber@hereon.de (M.N.); jan.bohlen@hereon.de (J.B.)

**Keywords:** magnesium, magnesium sheet, aluminum-free, magnesium zinc alloy, calcium-containing alloys

## Abstract

The processability during massive deformation of magnesium-wrought products is hampered by the low formability of magnesium alloys. The research results of recent years demonstrate that rare earth elements as alloying elements improve the properties of magnesium sheets, such as formability, strength and corrosion resistance. The substitution of rare earth elements by Ca in Mg-Zn-based alloys results in a similar texture evolution and mechanical behaviour as RE-containing alloys. This work is an approach to understanding the influence of Mn as an alloying element to increase the strength of a Mg-Zn-Ca alloy. For this aim, a Mg-Zn-Mn-Ca alloy is used to investigate how Mn affects the process parameters during rolling and the subsequent heat treatment. The microstructure, texture and mechanical properties of rolled sheets and heat treatment at different temperatures are compared. The outcome of casting and the thermo-mechanical treatment are used to discuss how to adapt the mechanical properties of magnesium alloy ZMX210. The alloy ZMX210 behaves very similarly to the ternary Mg-Zn-Ca alloys. The influence of the process parameter rolling temperature on the properties of the ZMX210 sheets was investigated. The rolling experiments show that the ZMX210 alloy has a relatively narrow process window.

## 1. Introduction

The low formability [1,2,3,4] of magnesium and its alloys also has a significant influence on all forming processes. The typical strong alignment of the basal planes in conventional magnesium sheets hinders the strain accommodation during deformation and the resulting work hardening ability [5]. The weaker textures of magnesium sheets lead mostly to a higher ductility, especially at room temperature, plus enhanced formability [1].

Rare earth elements containing magnesium alloys could partially compensate for drawbacks such as poor formability properties [1]. Magnesium sheet materials with the addition of rare earth elements show a randomized texture and fine-grained microstructure [5,6,7]. Thus, alloying rare earth elements affects a texture weakening characterized by a tilt of basal planes out of the sheet plane. To explain this behaviour, several approaches were used, which differ strongly from that of the same alloys without the addition of rare earth elements [8]. The mechanisms assumed to lead to such texture changes contain changes in the active deformation mechanisms, such as slip modes [5,8,9,10,11], different types of twins [6,8] and shear bands [12], and changes during microstructure regeneration as a result of recrystallization [13,14].

However, alloying with rare earth elements is difficult due to their strategic relevance for many industrial products and the strained supply situation. Therefore, they are categorized as a critical group of resources whose supply is combined with high costs, which induces economic dependence on imports of these elements. Furthermore, it was found that alloying of Ca to Mg–Zn alloys lead to textures that have strong similarities with those of rare earth elements containing Mg–Zn sheets [3,15,16].

This work will report on the effects of Mn in a Mg-Zn-based alloy with calcium as an additional alloying element with respect to the arising textures and sheet properties. The alloy ZMX210 will be exemplarily used to discuss the influence of the rolling temperature on the microstructure and texture development. According to the phase diagram of Mg-Zn-Mn-Ca-based alloys, these alloys have a eutectic with a low melting temperature and a Mg_6_Ca_2_Zn_3_ [17] phase forms at more than 0.2 wt% Ca. This eutectic has a melting point of a little above 400 °C, which only permits low forming temperatures due to the strong tendency to hot cracking.

## 2. Materials and Methods

The alloy ZMX210 was cast into slabs (120 mm × 105 mm, depth = 55–70 mm) by a modified gravity casting process. The composition of this alloy is listed in Table 1. To homogenize the microstructure and to decrease the number of precipitates, the billets were heat treated at 400 °C for 16 h after casting in an air atmosphere. For the rolling tests, slabs of the dimensions 100 mm × 150 mm × 20 mm were milled. All slabs were rolled to the final gauge on a Danieli rolling mill with roll dimensions of Ø 360 mm × 500 mm.

The slabs were rolled at 300 °C, 325 °C, 350 °C, 375 °C and 400 °C. The rolling procedure to the final gauge of approximately 1.8 mm consisted of 14 passes with different degrees of deformation (Table 2). φ is given as
φ = −ln(h_(n+1)_/h_n_),(1)
where n is the number of the pass and h_n_ is the sample thickness after pass n. Before the rolling procedure started, the slabs were heated for 30 min in an air atmosphere to the respective rolling temperature. The rolled samples were again reheated to the rolling temperature for 15 min before the following rolling passes. The sheets were air-cooled after the final rolling pass.

The water-soluble oil-based lubricant Oemeta SW 220 with a mix ratio of 1:100 was used, and the rolling speed was 10 m/min.

For recrystallization annealing after the rolling procedure, a part of the sheets was heat treated at the rolling temperature for 30 min in an air atmosphere. All sheets were cooled down by air. After all stages of sheet processing (as-rolled (AR) and heat-treated (HT)), the microstructures were investigated using optical microscopy. Standard metallographic specimen preparation procedures were used, and an etchant based on picric acid was used to make grains and grain boundaries visible [18]. To identify the composition of the matrix and the precipitates present in the alloy, as well as their composition changes with temperature and rolling process, energy dispersive X-ray spectroscopy (EDX) is employed with a scanning electron microscope (SEM: TESCAN VEGA3 and LYRA, Brno, Czechia). The texture was measured in the sheets on the sheet mid-planes using a Panalytical X-ray diffractometer setup and CuKα radiation. The first 6 reflections were used to measure pole figures up to an angle of 70° which allows recalculation of full pole figures using the open-source software version 5.1.1. routine MTEX [19]. The (0001) and (10-10) pole figures are shown in this work, presenting the sheet texture at midplane.

Local orientation patterns of single grains on the as-rolled and heat-treated sheets were measured by electron backscatter diffraction (EBSD) on a field emission gun scanning microscope (Zeiss, Ultra 55, EDAX/TSL, Oberkochen, Germany). The measurements were carried out on longitudinal sections of 200 µm × 550 µm in the centerline area of the sheets. An acceleration voltage of 15 kV and a step size of 0.4 μm was set. The sample surfaces were prepared in the same manner as for metallography and then electropolished with an AC2 solution (Struers™, Champigny-sur-Marne, France). The software “TSL Orientation Imaging Microscopy Analysis” version 7.1.1 x64 from EDAX© was employed to exploit the EBSD measurements. A cleansing function consists of a grain confidence index (CI) standardization, and a neighboring CI correlation was applied. The first function allows indexing measurement points correctly with low CI but a similar orientation to the surrounding measurement points. A grain tolerance angle of 5° and a minimum grain size of 2 points was defined. The second procedure was applied to measuring points with a CI lower than 0.1. In the case of a point with a lower CI, the neighbour with the highest confidence index is selected, and the orientation is substituted. The combination of both adjustment functions allowed a clear identification of the boundaries. EBSD data analysis included, in particular, a function for separating grains with different grain orientation spreads (GOS). A further restriction of a minimum grain size of 0.8 μm (again, 2 measured points) was set. With this approach, an average orientation was determined for a given grain. The deviation from the average orientation for each measuring point in this grain was then calculated and averaged as GOS. The software allowed the separation of grains with specified GOS.

In order to see how the different rolling temperatures and material conditions influence the mechanical properties of the sheets, tensile tests were conducted with a constant initial strain rate of 1.0 × 10^−3^ s^−1^. The tensile samples, DIN 50125—H 12.5 mm × 50 mm, of the as-rolled and heat-treated sheets were prepared in a rolling direction and transverse direction. All tensile tests were carried out on the universal tensile testing machine Zwick Z050 (Zwick, Ulm, Germany).

## 3. Results

### 3.1. Casting

Figure 1 presents the microstructures of the ZMX210 billets in the as-cast condition and after the heat treatment. The ZMX210 alloy reveals, in both conditions, a fine-grained microstructure with a homogeneous grain size distribution. A high amount of precipitates are also detected located at the grain boundaries, visible as bold black lines between grains, but also located in the grains. A heat treatment of 16 h at 400 °C decreases the number of precipitates at the grain boundaries (visible as thinner black lines between the grains) but does not significantly increase the average grain size.

EDX analyses were carried out for a more detailed examination of the microstructure and precipitates, see Figure 2. The EDX analysis of the cast material before heat treatment shows that little Zn, Mn and Ca are dissolved in the matrix (M1 and M2) and that these alloying elements are predominantly present as precipitates. The precipitates are present both linearly at the grain boundaries and globularly in the grains. The analysis of the elements at these points (P1–P4) shows that these precipitates contain relatively high amounts of Zn and Ca in addition to magnesium. This suggests that these precipitates are consistent with the Mg_6_Ca_2_Zn_3_ phases [17]. The Mn is mainly present as dot-shaped precipitates (P5) in the grains.

After the 16 h heat treatment at 400 °C, these phases partially disappear. The decreased amount of precipitates at the grain boundaries is explained by the fact that zinc dissolves in the matrix (M1, M2) mainly due to its solubility in magnesium (6.2 wt%, [20]). This is also reflected in the increase of the Zn content in the matrix (M1 and M2). The Ca particles (P4–P6) remain as Mg_6_Ca_2_Zn_3_ precipitates on the grain boundaries. The Mn precipitates (P1, P2) in the grains remain unaffected by the heat treatment. Because a reduction of the content of precipitates improves the forming behaviour, all slabs were heat-treated before rolling.

### 3.2. Rolling

The sheet rolled at 300 °C showed cold cracks after the 12th rolling pass. For the next rolling experiment, the temperature was increased to 325 °C, and the sheets were successfully rolled to a final gauge of 1.8 mm (Figure 3). Additionally, in the following rolling tests at 350 °C and 375 °C, the sheets could be rolled to the final thickness (Figure 3). However, as the temperature rises, the tendency to edge cracks increases sharply. The sheets rolled at 375 °C have strong edge cracks. In order to show the complete process window for the rolling procedure of ZMX210 sheets, the temperature was increased again to 400 °C. At a rolling temperature of 400 °C, hot cracks occur after the sixth rolling pass; thus, the rolling experiment was stopped at this point. So only between 325–350 °C is it possible to produce a sheet successfully with the applied rolling schedule.

Figure 4 shows the microstructures and textures of the sheets rolled at 325 °C, 350 °C and 375 °C. All sheets have a partially recrystallized microstructure with twinned grains. The results of the grain size measurement display that with increasing rolling temperature, the grain size increases slightly from 7 µm at 325 °C to 10 µm at 375 °C. The microstructure of the sheet rolled at 325 °C shows a high amount of small precipitates. These fine precipitates almost completely disappear with increasing rolling temperature. In addition to these very fine precipitates, linear precipitates and some recrystallized grains can be seen in all sheets.

Figure 4 also shows the pole figures of basal planes (0001) and prismatic planes (10-10) to display the texture of the as-rolled (AR) sheets. The textures of sheets of the three rolling temperatures are comparable in the as-rolled condition, i.e., the basal pole figures (0001) reveal a “donut”-shaped intensity distribution and a broad angular distribution of basal planes towards the transverse direction (TD), which will be reduced with elevated rolling temperature. The “donut”-shaped texture consists mainly of two texture components that dominate the crystallographic orientation. On the one hand, a component that is oriented in TD ({11-20}<10-10> component tilted 30° from TD to ND). On the other hand, a component is orientated in RD ({0001}<10-10> component tilted 30° from ND to RD). The prismatic pole figures (10-10) have the maximum in RD, which also decreases with increasing rolling temperature. The maximum intensity in RD belongs to the component/fiber tilted in TD.

The EDX analysis of the rolled sheets, see Figure 5, reveals that the precipitate stringers (highlighted in green) are consistent with the Mg_6_Ca_2_Zn_3_ phase. The Ca is mainly ligated in the Mg_6_Ca_2_Zn_3_ phases and is very present in the matrix. The comparatively large globular precipitates (highlighted in orange) in the grains are Mn particles. The very fine precipitates (highlighted in grey), which can be seen in the entire microstructure at a rolling temperature of 325 °C, are both Zn-based and Mn precipitates, as an EDX mapping with higher resolution shows (Figure 6, left side). Figure 6, right side, displays the EDX analyses for the sheet rolled at 375 °C, and it can be observed that the Zn precipitates dissolve in the matrix and the Mn particles stay. The Ca in the matrix is completely solved because even the EDX scan with the high resolution can clearly detect the Ca in the matrix.

For homogenization, all sheets received a 30 min heat treatment at the corresponding rolling temperature. After the heat treatment, the microstructures of all sheets are completely recrystallized (Figure 7). The heat treatment has no visible influence on the morphology of the precipitates. Here, the sheet rolled and heat-treated at 325 °C still has the most finely distributed precipitates. With increasing rolling and heat treatment temperatures, the number of precipitates is significantly reduced. The grain size does not increase as a result of the heat treatment—on the contrary, there is a very slight decrease in the average grain size because of recrystallization (see comparison to Figure 4).

After 30 min annealing of the rolled sheets, both the (0001) and (10-10) pole figures show changes (Figure 7). The basal pole figure (0001) shows only one double peak in the transverse direction ({11-20}<10-10> component tilted ~30° from TD to ND), which is slightly less pronounced if the temperature increases. In addition, the intensity of the prismatic pole figure somewhat decreases with increasing rolling temperature, which can be linked to the reduction of the broad angular distribution of basal planes towards TD.

Figure 8, Figure 9 and Figure 10 display the orientation patterns taken from EBSD measurements of all sheets in the as-rolled and in the heat-treated condition. The complete microstructure is shown in the first line, identified as “All Data”. In the next step, the microstructures are separated into two fractions by using the grain orientation spread (GOS) as a separator. It is supposed that fine grains with low orientation spread are the result of grain nucleation and growth mechanisms during recrystallization [21]. The specific limit of 1° is chosen arbitrarily to visualize only the recrystallized grains in the microstructure. Results are shown in all figures named “GOS < 1 recrystallized”. Similarly, a fraction of grains with a GOS value higher than 1° is separated to show the orientation of grains that are deformed but not recrystallized after the last rolling pass or heat treatment. These pictures are named “GOS > 1 deformed”. The EBSD results show a microstructure evolution consistent with the outcome descript in Figure 4 and Figure 7. After 30 min of heat treatment, all microstructures of the ZMX210 sheets are nearly fully recrystallized. The EBSD-measured local textures illustrate the same evolution as the X-ray-measured global textures. The basal pole figures (0001) with a “donut”-shaped intensity distribution and a broad angular distribution of basal planes towards the transverse direction (TD) change after the heat treatment to one double peak in the transverse direction. The EBSD measurements in the rolled condition clearly show that the prismatic fiber parallel to RD (<10-10>//RD) decreases with increasing rolling temperature.

### 3.3. Mechanical Properties

To determine the influence of process parameters and heat treatments on mechanical properties, uniaxial tensile tests were carried out. Figure 11 and Figure 12 show the stress–strain curves and mechanical properties of the sheets rolled at 325 °C and 350 °C at room temperature. The sheet rolled at 375 °C had too deep edge cracks so that no tensile specimens could be prepared. The diagrams (a) and (b) show the stress–strain curves of the sheets in the rolled condition and the diagrams (c) and (d) the curves for the sheets after heat treatment. In general, the stresses in the rolling direction are higher than in the transverse direction for both rolling temperatures and sheet conditions. On the other hand, mostly more than twice as high elongation values are achieved in the transverse direction, especially in the rolled state, than in the rolling direction.

The mechanical properties for the ZMX210 sheets directly after rolling do not differ significantly for the rolling temperatures of 325 °C and 350 °C. The ZMX210 sheets are available with a wide range of different mechanical properties. After heat treatment, the elongation values in the longitudinal and transverse directions are increased again by a factor of two. In addition, the stress–strain curves after heat treatment show a more pronounced yield strength. In particular, the curves of the specimens taken from the sheet in the transverse direction have an extremely pronounced yield strength. In contrast to all the other ZMX210 sheets investigated, the elongation values in the rolling direction are higher than in the transverse direction in the case of the sheet rolled and heat-treated at 325 °C. The elongation values of the ZMX210 sheet are also higher in the longitudinal direction than in the transverse direction. The sheets that were rolled and heat treated at 350 °C achieved the highest elongation values. The strength of the sheets after heat treatment decreases moderately.

## 4. Discussion

The results of this work clearly show that the quaternary alloy ZMX210 behaves very similarly to the ternary Mg-Zn-Ca alloys [15]. The rolling trials show that the ZMX210 alloy has a relatively narrow process window. Below a rolling temperature of 325 °C, cold cracks occur, and above 400 °C, hot cracks lead to an end of the rolling tests. Even at 375 °C, strong hot cracks at the sheet edges were observed (Figure 3). The reason for the hot cracks is the presence of the Mg_6_Ca_2_Zn_3_ phase in the sheet. The melting point of this phase is around 400 °C, so the additional heat coming from the rolling leads to hot cracks [22].

The variation of the rolling temperature has a relatively small but visible influence on the microstructure in the sheets with regard to grain size and grain size distribution. A fine-grained, homogeneous microstructure can be achieved with all three rolling temperatures. However, the rolling temperature has a large influence on the precipitates in the microstructure of the sheets. The increase in the rolling temperature leads to a decrease in the precipitates in the matrix; because of the higher rolling temperature, the Zn, in particular, dissolves better in the matrix. This can be seen in the high-resolution EDX scans in Figure 6.

The textures of the investigated sheets show the typical behaviour of rare-earth-containing Mg-Zn alloys with a distinct influence on the rolling temperature and subsequent heat treatment. The basal pole figures (0001) of the rolled sheets have a donut-shaped orientation, and the pole figures (10-10) have a maximum in the rolling direction before heat treatment. After the heat treatment, both the (0001) and (10-10) pole figures show changes (Figure 7). The basal pole figure (0001) shows only one double peak in the transverse direction, which is slightly less pronounced if the temperature increases. Results of previous work show the correlation of specific texture components in magnesium sheets to specific deformation mechanisms and to preferential orientation growth during recrystallization. According to Styczynski [9], the split peaks in the rolling direction in AZ31 are the result of <c + a> slip. The transverse spread has been associated with an enhanced prismatic slip if deformation dominates the texture development [23]. Bohlen et al. [15] have shown that the transverse spread of poles is stabilized into two single-intensity peaks as a result of recrystallization during annealing. The texture development is changed due to a higher impact of the underlying recrystallization mechanism as the temperature increases. Another hind for this behaviour is the increase of grain size with increasing rolling temperature. One more reason for such texture development is a different deformation behaviour, especially the formation of different types of twins during rolling [15].

Figure 13 shows for the ZMX210 sheets rolled at 325 °C and 375 °C, the boundary maps with twins (tensile red, compression blue and double yellow) and misorientation angle distribution, which shows the fraction of different twin modes in the microstructure. It becomes clear that at the higher rolling temperature of 375 °C, the percentage of double and tensile twins is higher than at the lower rolling temperature (325 °C). Instead, more compression twins can be seen in the microstructure at lower rolling temperatures. The higher number of tensile twins in the (0001) pole figure leads to a stronger formation of the texture component in the rolling direction after the final heat treatment. The formation of different twins is caused, on the one hand, by the different rolling temperatures, which activate different deformation mechanisms [24]. On the other hand, this can also be because of the different distribution of the elements in the matrix. Although the EDX scans in Figure 5 shows an almost identical element distribution in the matrix for both rolling temperatures, the EDX scans with increased resolution (Figure 6) show that the Zn occurs as larger precipitates in the matrix at a rolling temperature of 325 °C. In the sheet rolled at 375 °C, the Zn is very finely distributed in the matrix.

In the tensile test, the sheets in the rolled state show relatively low elongation values, which indicates poor forming properties. In particular, the stress–strain curves in the rolling direction show no hardening at all, which suggests strong hardening in the sheet. The stress–strain curves in the transverse direction show behaviour that is much more ductile. This is an effect of the texture that occurs after rolling, which prefers basal conduction in the transverse direction. In the as-rolled state, the process temperature during rolling has no influence on the mechanical properties. 

After the heat treatment of the sheets, the elongation values of the sheets in the rolling and transverse directions increase significantly. This can be explained by the complete recrystallization of the microstructure. The elongation values and strength in the rolling direction as a function of the rolling temperature are close to each other. The tensile tests at 325 °C in the transverse direction lead to premature failure, which can be attributed to the increased number of small particles in the microstructure.

## 5. Conclusions

Since the aim is to achieve as isotropic a deformation behaviour as possible in sheet metal, the ZMX210 alloy should be rolled at higher temperatures. This is because, at this rolling temperature, a texture with a lower split in TR occurs, which favours isotropic forming behaviour. However, the higher rolling temperatures are countered by the increased tendency to edge cracks.

## Figures and Tables

**Figure 1 materials-16-04227-f001:**
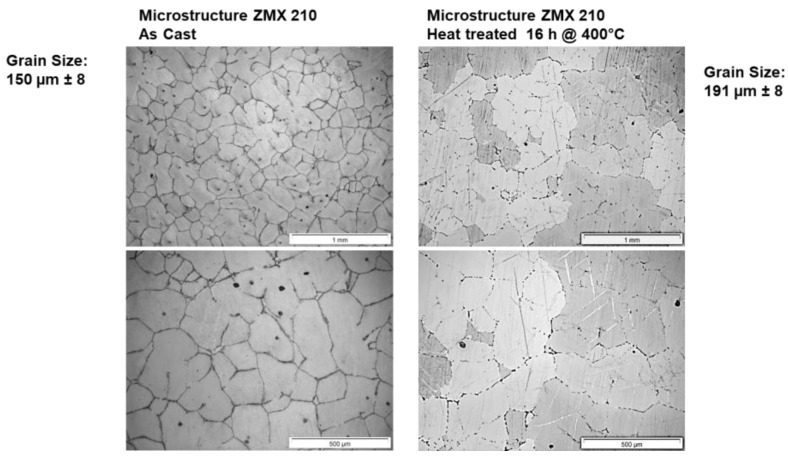
Microstructure of the alloy ZMX210 in as-cast and heat-treated conditions (two different magnifications).

**Figure 2 materials-16-04227-f002:**
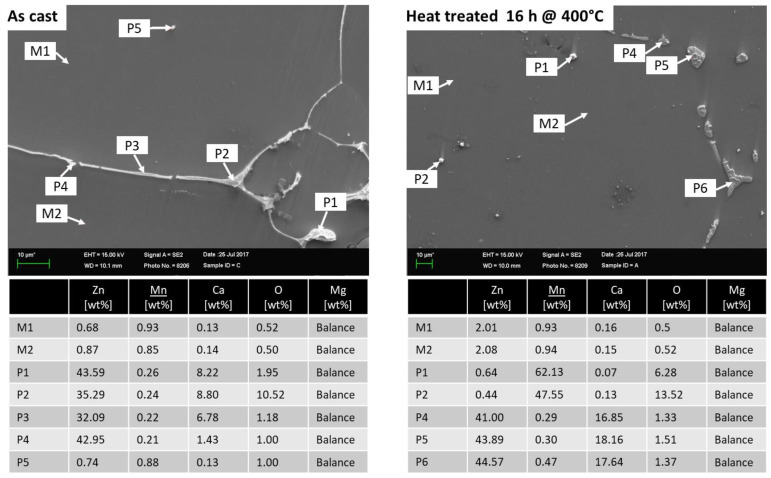
EDX analyses of the rolling slabs in as-cast and heat-treated conditions.

**Figure 3 materials-16-04227-f003:**
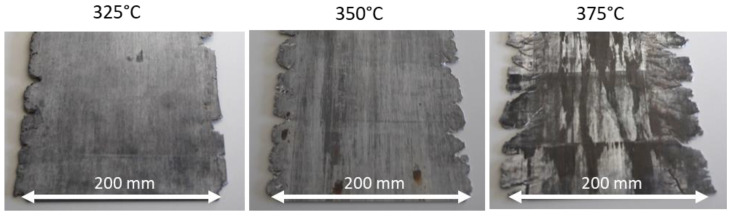
ZMX210 sheets rolled at different temperatures.

**Figure 4 materials-16-04227-f004:**
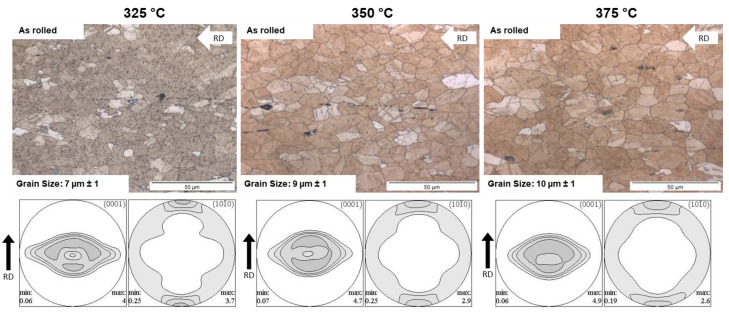
Microstructure and texture of the ZMX210 sheets in the as-rolled condition.

**Figure 5 materials-16-04227-f005:**
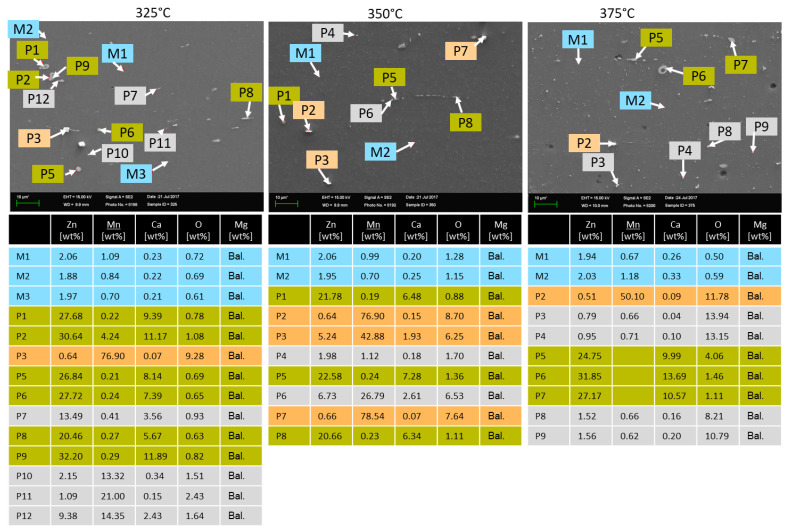
EDX analyses of the as-rolled sheets.

**Figure 6 materials-16-04227-f006:**
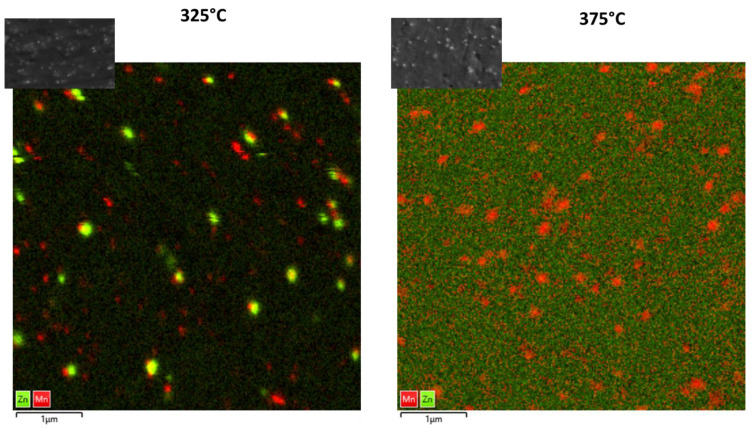
EDX-Mapping at higher resolution (83.3 k×).

**Figure 7 materials-16-04227-f007:**
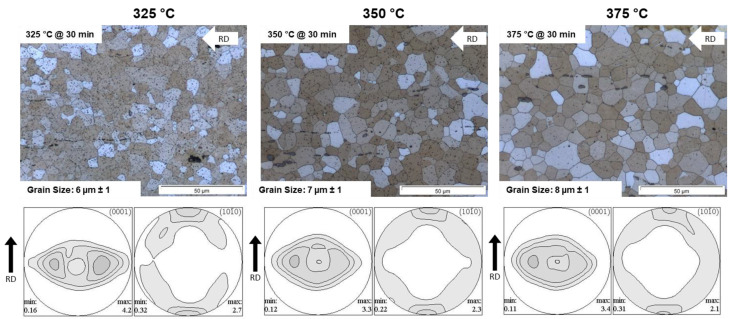
Microstructure and texture of the ZMX210 sheets after heat treatment.

**Figure 8 materials-16-04227-f008:**
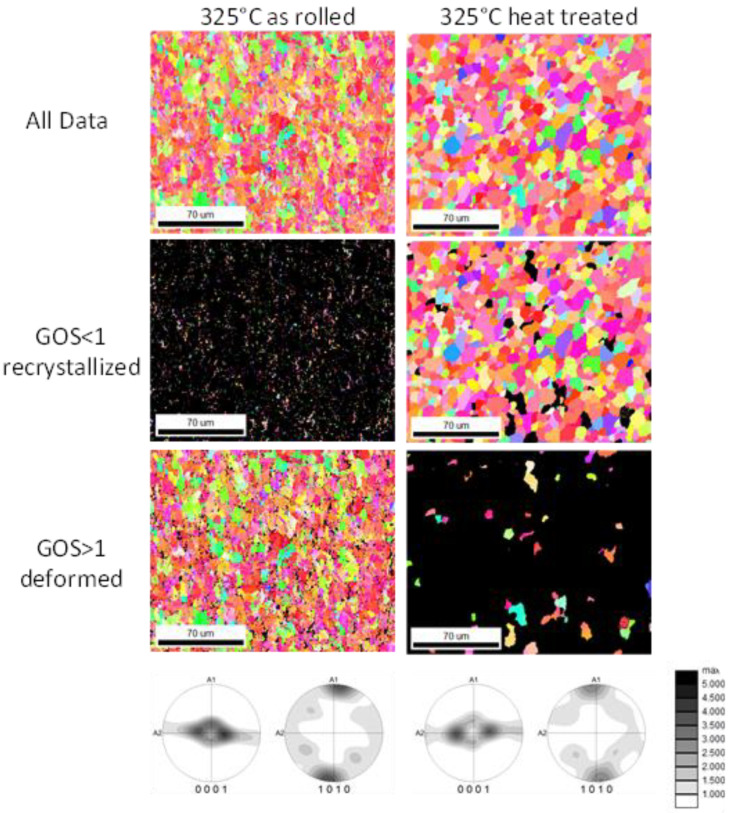
Orientation maps of the ZMX210 sheets rolled at 325 °C with GOS separation.

**Figure 9 materials-16-04227-f009:**
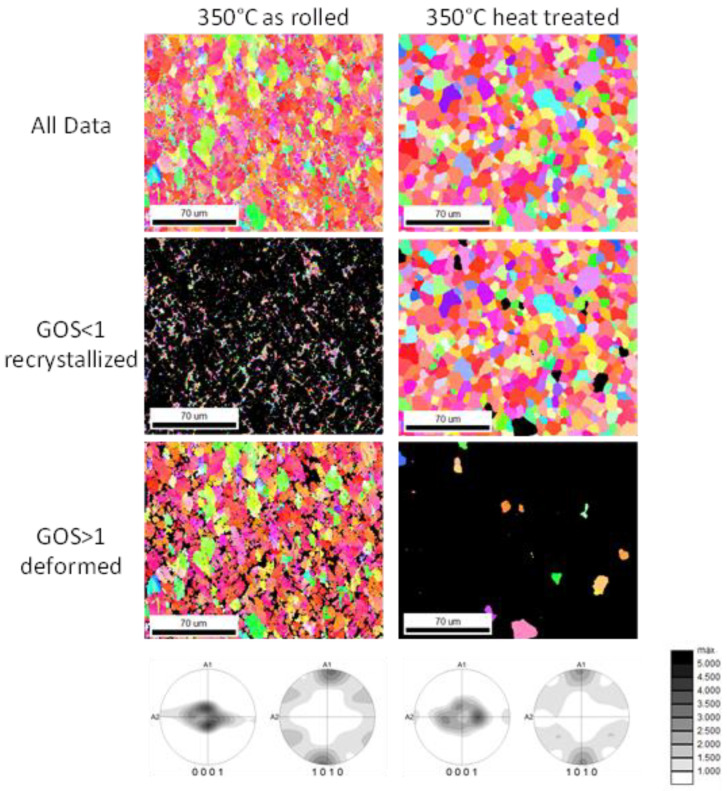
Orientation maps of the ZMX210 sheets rolled at 350 °C with GOS separation.

**Figure 10 materials-16-04227-f010:**
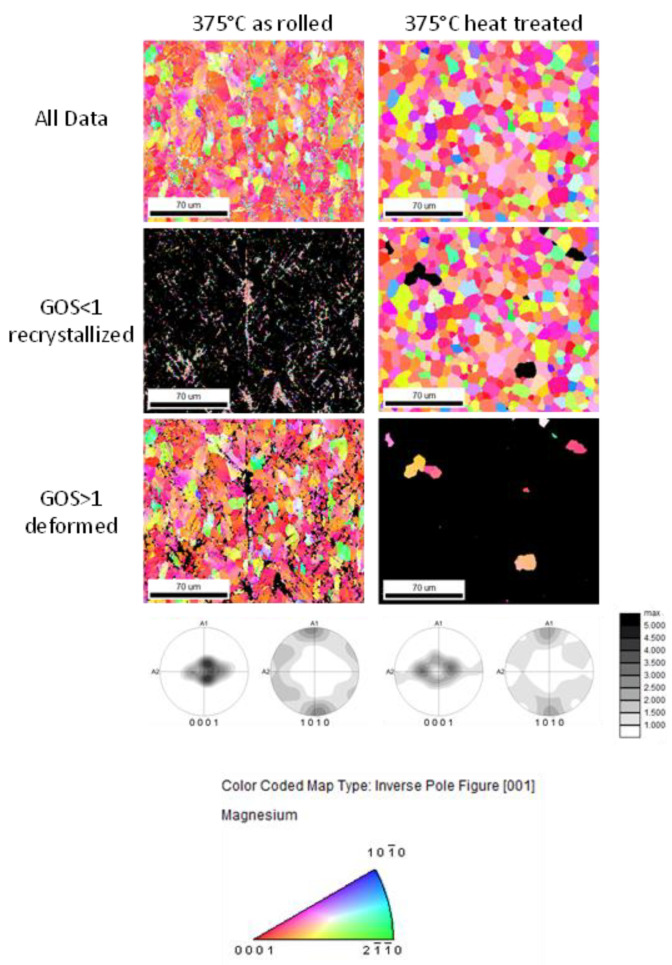
Orientation maps of the ZMX210 sheets rolled at 375 °C with GOS separation.

**Figure 11 materials-16-04227-f011:**
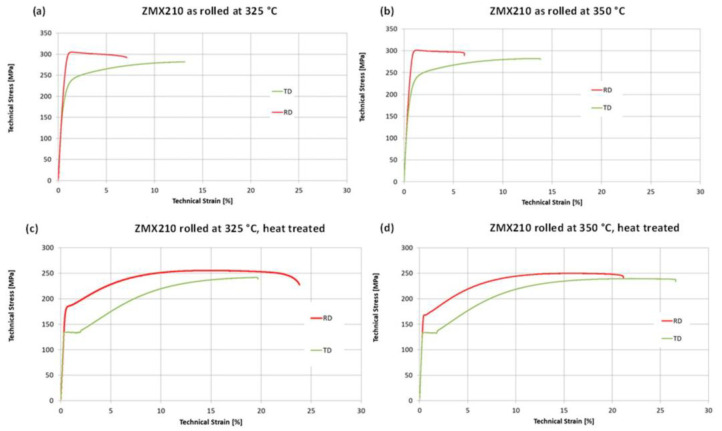
Stress–strain diagrams of the sheets rolled at 325 °C and 350 °C in as-rolled and heat-treated condition.

**Figure 12 materials-16-04227-f012:**
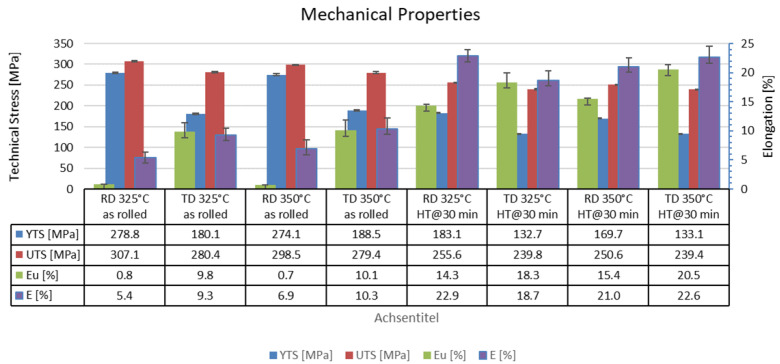
Mechanical properties of the sheets rolled at 325 °C and 350 °C in as-rolled and heat-treated conditions.

**Figure 13 materials-16-04227-f013:**
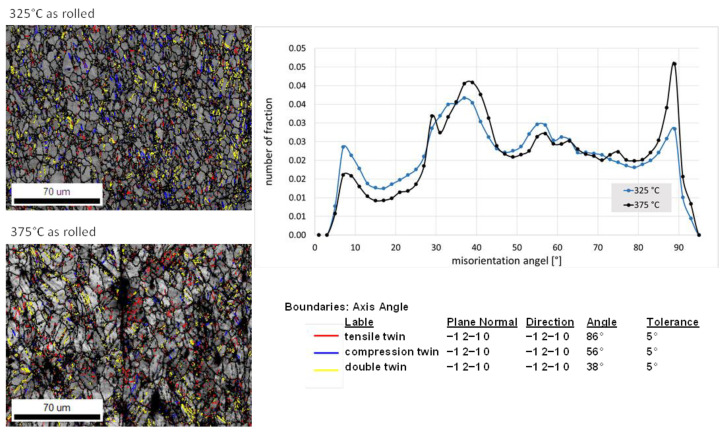
Boundary maps with twins of the ZMX210 sheets in the as-rolled condition (325 °C and 375 °C) and the number of fractions of the twins; black: grain boundary, red: “tensile twins”, blue: “compression twins”, yellow: “double twins”.

**Table 1 materials-16-04227-t001:** Alloy composition in wt% (mg in balance).

	Zn	Mn	Ca	Fe	Cu	Ni	Si
Mg2Zn1Mn0.3Ca	2.30	0.91	0.26	0.0025	0.0009	0.0021	0.016

**Table 2 materials-16-04227-t002:** Rolling schedules.

φ	Planned Thickness	300	325	350	375	400	Heating Time (min)
	20.00						30
0.1	18.10	18.82	18.30	18.24	18.22	18.24	10
0.1	16.37	16.52	16.52	16.52	16.52	16.50	10
0.1	14.82	14.92	14.96	14.92	14.90	14.96	10
0.1	13.41	13.50	13.48	13.38	13.40	13.36	10
0.2	10.98	12.22	11.20	11.14	11.10	11.06	10
0.2	8.99	11.06	9.14	9.06	9.06	cracks	10
0.2	7.36	9.08	7.44	7.30	7.32		10
0.2	6.02	7.46	6.08	5.96	5.98		10
0.2	4.93	6.10	5.06	4.98	4.98		10
0.2	4.04	5.90	4.00	3.90	3.92		10
0.2	3.31	5.30	3.40	3.30	3.32		10
0.2	2.71	cracks	2.80	2.70	2.78		10
0.2	2.22		2.28	2.22	2.32		10
0.2	1.81		1.80	1.82	1.86		10

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
