# Peer review of "Microstructure and Texture Evolution of the Magnesium Alloy ZMX210 during Rolling and Annealing"

_materials, 2023, doi:10.3390/ma16124227_

Round 1

Reviewer 1 Report

The manuscript reports on the microstructure and text evolution of Mg-based ZMX210 by rolling and annealing to understand the influence of the process parameters and the influence of alloying elements. There are some omissions in result and description. Therefore it is worth publishing this manuscript provided that the following concerns are addressed properly.

#1. The novelty of the paper needs to be emphasized distinctly in the Introduction. Abstract does not supports the essential of this paper. The summary of quantitative result needs to be added in the Abstract.

#2. Fig.1 just shows the equipment without any meaningful information for experiment such as details of equipment, appearance of specimen during processing, etc.. Therefore, it is considered excessive and unnecessary data.

#3. Please provide the more detailed processing conditions such as atmosphere (in air or other gas) cooling condition (in air or furnace or rapid quenching) for annealing processing on homogenization and recrystallization.

#4. For Fig. 2, it has been mentioned that “The heat treatment of 16 h at 400 °C decreases the number of precipitates at the grain boundaries,” The black dots on the grain boundary of heat treated specimen in Fig.2 is to be considered precipitate. Is it right?

#5. For Fig. 3, it is hard to read the index of points. It need to improve the readability. For EDX result (table) of heat treated specimen in Fig. 3, the total composition on P1, P2, and P3 points has to be 100 %.

What is the “*” at P3* point? Why 55.37 wt.%Cu content was detected?

#6. EDX results in Fig. 5 must be in 100 % in total composition for each point.

#7. For 3.1. Mechanical Properties, please provide the quantitative values of mechanical properties, such as yield strength, UTS, and plasticity.

It need to moderate editing of English language.

Author Response

  1. The novelty of the paper needs to be emphasized distinctly in the Introduction. Abstract does not supports the essential of this paper. The summary of quantitative result needs to be added in the Abstract.

Response 1.: The abstract was changed accordingly.

2. Fig.1 just shows the equipment without any meaningful information for experiment such as details of equipment, appearance of specimen during processing, etc.. Therefore, it is considered excessive and unnecessary data.

Response 2.: Figure 1 was deleted.

3. Please provide the more detailed processing conditions such as atmosphere (in air or other gas) cooling condition (in air or furnace or rapid quenching) for annealing processing on homogenization and recrystallization. 

Response 3.: The text has been modified to include the relevant information.

4. For Fig. 2, it has been mentioned that “The heat treatment of 16 h at 400 °C decreases the number of precipitates at the grain boundaries,” The black dots on the grain boundary of heat treated specimen in Fig.2 is to be considered precipitate. Is it right?

Response 4.: That is right. The text was changed to clarify this.

5. For Fig. 3, it is hard to read the index of points. It need to improve the readability. For EDX result (table) of heat treated specimen in Fig. 3, the total composition on P1, P2, and P3 points has to be 100 %.
What is the “*” at P3* point? Why 55.37 wt.%Cu content was detected?

Response 5.: Figure 3 the EDX images and tables were modified accordingly. The Cu detected in P3 is impurities originating either from the casting process or the sample preparation. Therefore, the data point P3 was deleted.

6. EDX results in Fig. 5 must be in 100 % in total composition for each point.

Response 6.: The EDX tables were modified accordingly.

7. For 3.1. Mechanical Properties, please provide the quantitative values of mechanical properties, such as yield strength, UTS, and plasticity.

Response 7.: A table of the mechanical properties is added.

Reviewer 2 Report

This work focused on the microstructure and texture evolution of the magnesium alloy ZMX210 during rolling and annealing. The topic is meaningful and some good results were obtained. However, the authors should address the following issues before this paper is accepted for publication in Materials.

(1) Kindly check and revise the typo errors, such as superscript and indices of crystal face.

(2) The authors mentioned that rolling temperature has a significant impact on the surface forming quality of ZMX210 alloy. Can you provide corresponding macroscopic photos to illustrate this.

(3) What are the specific texture components for texture analysis? ODF analysis can be considered.

(4) The conclusion and discussion sections of the paper should be separated.

This work focused on the microstructure and texture evolution of the magnesium alloy ZMX210 during rolling and annealing. The topic is meaningful and some good results were obtained. However, the authors should address the following issues before this paper is accepted for publication in Materials.

(1) Kindly check and revise the typo errors, such as superscript and indices of crystal face.

(2) The authors mentioned that rolling temperature has a significant impact on the surface forming quality of ZMX210 alloy. Can you provide corresponding macroscopic photos to illustrate this.

(3) What are the specific texture components for texture analysis? ODF analysis can be considered.

(4) The conclusion and discussion sections of the paper should be separated.

Author Response

(1) Kindly check and revise the typo errors, such as superscript and indices of crystal face.

Response 1: The typos were eliminated, especially the superscript and indices of crystal face.

(2) The authors mentioned that rolling temperature has a significant impact on the surface forming quality of ZMX210 alloy. Can you provide corresponding macroscopic photos to illustrate this.

Response 2: A photo was added.

(3) What are the specific texture components for texture analysis? ODF analysis can be considered.

Response 3: The specific texture components for texture analysis were added.

(4) The conclusion and discussion sections of the paper should be separated.

Response 4: The conclusion and discussion sections were separeted.

Round 2

Reviewer 1 Report

Authors have revised manuscript according to the comments arised by reviewer. Therefore, the revised manuscript is recommended for publication in Materials.